# SpecRaGE: Robust and Generalizable Multi-view Spectral Representation Learning

## Abstract

Multi-view representation learning (MvRL) has garnered substantial attention in recent years, driven by the increasing demand for applications that can effectively process and analyze data from multiple sources. In this context, graph Laplacian-based MvRL methods have demonstrated remarkable success in representing multi-view data. However, these methods often struggle with generalization to new data and face challenges with scalability. Moreover, in many practical scenarios, multi-view data is contaminated by noise or outliers. In such cases, modern deep-learning-based MvRL approaches that rely on alignment or contrastive objectives can lead to misleading results, as they may impose incorrect consistency between clear and corrupted data sources. We introduce *SpecRaGE*, a novel fusion-based framework that integrates the strengths of graph Laplacian methods with the power of deep learning to overcome these challenges. SpecRage uses neural networks to learn parametric mapping that approximates a joint diagonalization of graph Laplacians. This solution bypasses the need for alignment while enabling generalizable and scalable learning of informative and meaningful representations. Moreover, it incorporates a meta-learning fusion module that dynamically adapts to data quality, ensuring robustness against outliers and noisy views. Our extensive experiments demonstrate that SpecRaGE outperforms state-of-the-art methods, particularly in scenarios with data contamination, paving the way for more reliable and efficient multi-view learning. Our code will be made publicly available upon acceptance.

## 1 Introduction

Multi-view representation learning (MvRL) has become a crucial paradigm in recent years. Its primary objective is to integrate data from multiple sources into a unified representation, which can be used for various tasks such as clustering and classification. The demand for MvRL methods has surged as more applications require analyzing objects or phenomena from diverse perspectives. For instance, streaming platforms rely on the fusion of visual, audio, and textual features to enhance content recommendations, while healthcare systems combine genetic data, imaging, and clinical records to provide a comprehensive view of patient health.

Graph Laplacian methods have demonstrated notable effectiveness in representing both single-view and multi-view data (Ng et al., 2001; Belkin & Niyogi, 2003; Coifman & Lafon, 2006a; Cai et al., 2011; Kumar et al., 2011; Eynard et al., 2012; 2015). This success largely stems from the ability of Laplacian eigenvectors to preserve similarity and capture the underlying cluster structure of the data (Ng et al., 2001; Belkin & Niyogi, 2003). Moreover, by focusing on the eigenvectors corresponding to the smallest eigenvalues, one can uncover the intrinsic low-dimensional manifold structure, as demonstrated in *Laplacian eigenmaps* (Belkin & Niyogi, 2003).

However, in the multi-view setting, traditional graph Laplacian-based methods face two major limitations: *Generalizability* - the ability to map new test points into the representation space after the training set has been processed (i.e., out-of-sample extension); *Scalability* – the capability to process large datasets within a practical time frame efficiently. Current graph Laplacian-based MvRL approaches often fail to address these two challenges (see discussion in Section 2). These limitations in generalizability and scalability hinder the full potential of graph Laplacians for representing multi-view data in real-world applications.

Another fundamental challenge in the multi-view setting is the presence of noise or outliers in some of the views. For example, in autonomous driving systems that integrate data from multiple sensors, adverse weather conditions can lead to unreliable or noisy data from certain sensors. Most modern deep learning-based MvRL approaches (Andrew et al., 2013; Huang et al., 2019; 2021; Federici et al., 2020; Trosten et al., 2021; Xu et al., 2022; Trosten et al., 2023; Wang et al., 2023; Yan et al., 2023) rely on alignment or contrastive objectives that enforces consistency between view-specific representations. These alignment-based methods can become problematic when handling contaminated data, as they may enforce incorrect consistency between clear and degraded data sources. These challenges highlight the pressing need for robust MvRL methods that can effectively handle real-world data imperfections while maintaining the benefits of graph Laplacian-based approaches.

To address the challenges of *Generalizability* and *Scalability* in graph Laplacian-based methods, as well as ensuring *Robustness* in the presence of contaminated views, we propose *SpecRaGE*, a novel fusion-based MvRL framework that combines the strengths of graph Laplacians with the power of deep learning. At its core, SpecRaGE provides a deep-learning solution to the joint diagonalization of Laplacians, by extending *SpectralNet* (Shaham et al., 2018) to multi-view settings. This joint diagonalization is our key strategy for avoiding the need for alignment between views (see Section 4.1 for further discussion). SpecRaGE is inherently scalable, as it is trained on mini-batches in a stochastic manner, allowing it to efficiently process large datasets. Furthermore, the resulting model provides a parametric map that approximates the leading joint eigenvectors of multi-view graph Laplacians. This parametric map enables efficient application to new data, addressing the generalizability challenge of traditional graph Laplacian methods.

Moreover, SpecRaGE incorporates a flexible fusion technique that overcomes the rigid limitations of traditional alignment-based methods when dealing with contaminated multi-view data. Specifically, SpecRaGE introduces a meta-learning-driven fusion mechanism, which generates sample-specific weight vectors, allowing the model to down-weight anomalous or noisy views dynamically.

Our extensive experiments demonstrate that SpecRaGE not only achieves state-of-the-art performance on standard multi-view benchmarks but also significantly outperforms existing methods when dealing with outliers and incomplete views.

The main contributions of this work are: (1) We introduce a generalizable and scalable, graph Laplacian-based MvRL framework that extends the power of spectral methods to large-scale multi-view data. (2) We propose a meta-learning fusion module that dynamically weights different views, providing robust performance in the face of data contamination. (3) We present extensive experimental results demonstrating SpecRaGE's superior performance across various benchmarks, particularly in scenarios involving outliers and noisy views.

## 2 RELATED WORK

**Graph Laplacian-based Methods.** Various graph Laplacian-based methods (also known as multi-view spectral representation learning methods) have been proposed to extract compact and informative representations from multi-view data (Cai et al., 2011; Kumar et al., 2011; Eynard et al., 2012; 2015; Li et al., 2015; Lindenbaum et al., 2020; Yang et al., 2023). These approaches typically aim to learn a fused representation based on multiple graph Laplacians, one for each view. While these methods produce meaningful representations, they often face challenges with generalizability (out-of-sample extension), requiring the recomputation of graph Laplacians to embed new, unseen data into the fused representation space. Some approaches address this issue through out-of-sample extension methods, such as the Nystrom extension (Nyström, 1930) or Geometric Harmonics (Coifman & Lafon, 2006b). However, these techniques were originally developed for single-view data and provide only local extensions, functioning effectively only near existing training points. Additionally, they are computationally intensive and memory-demanding, as they require calculating distances between each new test point and all training points. Furthermore, many graph Laplacian-based MvRL approaches struggle with scalability due to their high computational demands and limited support for sparse matrices, complicating the storage and processing of large Laplacians. These issues with generalizability and scalability hinder the practical application of graph Laplacians for representing multi-view data in real-world settings.

**Deep-learning based Methods.** Modern deep-learning-based MvRL methods attempt to design a loss function that is useful for extracting meaningful representations from multi-view data. One category of methods includes deep extensions of the Canonical Correlation Analysis (CCA) algorithm, such as DCCA, DCCAE, and $\ell_0$-DCCA (Hotelling, 1936; Andrew et al., 2013; Wang et al., 2015; Lindenbaum et al., 2021). These methods utilize deep networks to learn a non-linear mapping for two views, maximizing their correlations. Another set of algorithms relies on information-theory-based metrics (Federici et al., 2020; Lin et al., 2022), aiming to maximize the mutual information between views while minimizing redundant information unique to each view. Contrastive learning methods (Trosten et al., 2021; Xu et al., 2022; Yan et al., 2023; Wang et al., 2023) represent another group of deep learning approaches, utilizing a contrastive alignment objective to achieve view-specific alignment. Deep learning techniques have also been applied to address the multi-view spectral clustering problem (Huang et al., 2019; 2021). These methods are closely related to our work, as they also extend SpectralNet to multi-view settings by incorporating alignment objectives between the spectral embeddings from each view. Despite their promising performance, these approaches rely on some form of alignment between the view-specific representations, making them vulnerable to data contamination. Their underlying assumptions regarding data quality and view consistency may struggle in the presence of noise, outliers, or asymmetric corruption across views, as demonstrated in Section 5.3.

## 3 PRELIMINARIES

### 3.1 GRAPH LAPLACIAN AND SPECTRALNET

**Graph Laplacian.** Given a dataset of $n$ points $x_1, x_2, \ldots, x_n$, a positive semi-definite and symmetric affinity matrix $W$ is an $n \times n$ matrix whose $W_{i,j}$ entry represents the similarity between $x_i$ and $x_j$. The unnormalized graph Laplacian is defined as $L = D - W$ where $D$ is a diagonal matrix in which the element $D_{i,i} = \sum_{j=1}^{n} W_{i,j}$ correspond to the degrees of the points $x_i$, $i = 1, ..., n$. The eigenvectors corresponding to the smallest eigenvalues of the Laplacian provide valuable low-dimensional representations, capturing structural information like relationships and similarities between data points. These eigenvectors are widely used in applications such as spectral clustering (Ng et al., 2001), dimensionality reduction (Belkin & Niyogi, 2003), graph partitioning (Karypis & Kumar, 1998), and image segmentation (Shi & Malik, 2000; Melas-Kyriazi et al., 2022).

**SpectralNet.** *SpectralNet* (Shaham et al., 2018) is a deep-learning model that effectively maps single-view data to the approximate eigenvectors of its Laplacian. This enables the performance of spectral clustering on huge single-view datasets since the loss is amenable to parallelized training. Furthermore, the model can be easily and accurately used to generalize the representation to unseen test data. To learn the eigenvectors, *SpectralNet* minimizes the following Rayleigh-quotient loss: $\text{Tr}\left(Y^T L Y\right)$ s.t. $Y^T Y = I$, where $Y \in \mathbb{R}^{n \times k}$ is the network's output and $L$ is the graph Laplacian.

### 3.2 JOINT DIAGONALIZATION

**Definition 1.** *A set of diagonalizable matrices $A^{(1)}, A^{(2)}, \ldots, A^{(v)}$ is said to be simultaneously diagonalizable if there exists a single invertible matrix $V$ such that for all $1 \leq p \leq v$, $V^{-1} A^{(p)} V$ is a diagonal matrix.*

Intuitively, *joint diagonalization* (or *simultaneous diagonalization*) seeks to find a common basis by which all matrices could be represented in a diagonal form.

However, exact joint diagonalization can be achieved if and only if $A^{(1)}, A^{(2)}, \ldots, A^{(v)}$ commute. Nevertheless, when commutativity cannot be guaranteed, optimizing a joint diagonality criterion and approximating the solution remains possible. This defines the *approximate joint diagonalization* problem.

**Approximate Joint Diagonalization of Laplacians.** In terms of graph Laplacians, for a set of graph Laplacians $L^{(1)}, L^{(2)}, \ldots, L^{(v)}$ the objective of *joint diagonalization* is to find a set of orthogonal eigenvectors $V$ such that for each Laplacian $L^{(p)}$, the matrix $V^T L^{(p)} V$ is diagonal and contains the eigenvalues of $L^{(p)}$ on the diagonal. It has been demonstrated in (Eynard et al., 2012; 2015), that

it is possible to find an approximate joint diagonalization of Laplacians by solving the following objective:

$$\min_{V \in \mathbb{R}^{n \times k}} \text{Tr}\left(V^T \bar{L} V\right), \quad \text{s.t.} \quad V^T V = I, \tag{1}$$

where $\bar{L}$ represents a form of average of the graph Laplacians. This average can be computed, for example, as a weighted arithmetic mean: $\bar{L} = \sum_{p=1}^{v} w^{(p)} L^{(p)}$ where $w^{(p)}$ represent the contribution of the $p$-th view.

## 4 SPECRAGE

**Problem Statement.** Let $X = \{\mathcal{X}^{(p)} \in \mathbb{R}^{n \times d_p}\}_{p=1}^{v}$ be a multi-view dataset where $n$ is the number of samples, $v$ is the number of views, and $d_p$ denotes the dimensionality of samples within the $p$-th view. MvRL aims to leverage the multi-view information to learn a high-quality unified representation that facilitates downstream tasks such as clustering, manifold learning, or classification.

In this section, we introduce SpecRaGE, our MvRL framework designed to tackle the challenges of generalizability, scalability, and robustness to contaminated views. We begin by outlining the rationale behind our method. Next, we detail how SpecRaGE efficiently learns the approximate joint eigenvectors, facilitating generalization to new data. Finally, we explore the meta-learning fusion module, which enhances robustness in the presence of contaminated views. The overall framework of our method is illustrated in Fig. 1.

### 4.1 RATIONAL

To develop a method that effectively harnesses the compelling properties of graph Laplacians while enhancing robustness against data contamination, it is essential to first eliminate the need for alignment and then implement a mechanism that diminishes the influence of low-quality views. To achieve this, SpecRaGE utilizes joint diagonalization of Laplacians, which fuses information from all views into a unified representation that approximates the joint eigenvectors. By prioritizing the fusion of information over the alignment of view representations, we successfully circumvent the challenges associated with rigid view alignment. Furthermore, to enhance robustness, SpecRaGE incorporates a dynamic fusion module that adjusts the weighting of views based on their quality. This effectively reduces the influence of noisy or outlier views. This fusion-based approach not only leverages the advantages of graph Laplacians but also demonstrates resilience to contaminated data. Crucially, we approximate these joint eigenvectors in a scalable manner using deep learning, facilitating efficient processing of large datasets and enabling generalization to new, unseen data.

### 4.2 GENERALIZABLE AND SCALABLE JOINT EIGENVECTORS APPROXIMATION

Let $F_\theta : \mathbb{R}^{d_1} \times \mathbb{R}^{d_2} \times \cdots \times \mathbb{R}^{d_v} \to \mathbb{R}^k$ be a parametric mapping (i.e., a neural network model) that transforms a multi-view input into the corresponding coordinates in a fused representation. That is, for a multi-view input $\hat{x}_i = \left(x_i^{(1)}, x_i^{(2)}, \ldots, x_i^{(v)}\right)$, $F_\theta$ operates such that $y_i = F_\theta\left(\hat{x}_i\right)$ where $y_i \in \mathbb{R}^k$ is the corresponding coordinates in the fused representation. To perform a deep joint diagonalization of Laplacians for a batch of $m$ multi-view data points, we propose the following loss:

$$\mathcal{L}_\theta = \frac{1}{m^2 v} \sum_{p=1}^{v} \sum_{i,j=1}^{m} W_{i,j}^{(p)} \|y_i - y_j\|_2^2, \tag{2}$$

where $W^{(p)}$ is an $m \times m$ affinity matrix of the $p$-th view, and $W_{i,j}^{(p)}$ represents an affinity measure between $x_i^{(p)}$ and $x_j^{(p)}$. This loss function encourages similar points (as measured by the affinity matrices) to be close in the fused representation space. As one can observe, this loss can be minimized by mapping all points to the same output (e.g., $F_\theta(\hat{x}_i) = y_0$ for all $i$). To avoid such a trivial map, an orthogonality constraint is added to the output, i.e.,

$$Y^T Y = I_{k \times k}, \tag{3}$$

where $Y$ is an $m \times k$ matrix of the outputs whose $i$-th row is $y_i$.

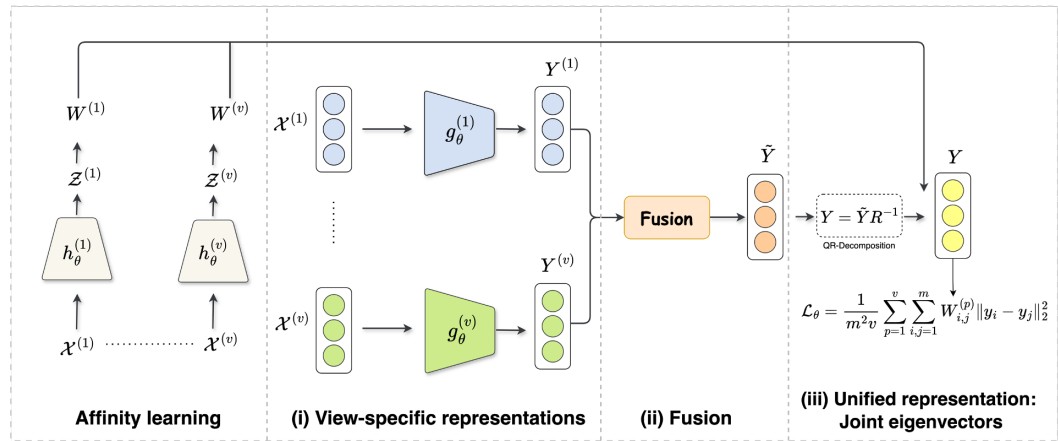

Figure 1: SpecRaGE architecture. First, $v$ view-specific representations are obtained from $v$ corresponding neural networks. Subsequently, a fusion is performed. Following the fusion, the fused representation undergoes QR decomposition to ensure orthogonality. Finally, the loss function (Eq. 2 or Eq. 5) is computed where the affinity matrices are constructed by the embeddings of the pre-trained Siamese networks (see discussion in Appendix B). All networks weights are updated by the gradients.

To satisfy the orthogonality constraint, we follow the same technique from *SpectralNet* and construct an orthogonalization layer that computes the QR decomposition of the network output and returns the orthogonal $Q$ matrix. More specifically, let $\tilde{Y}$ denote the $m \times k$ matrix obtained from the fusion step; the weights of this layer are defined to be the matrix $R^{-1}$ from the QR decomposition of $\tilde{Y}$. The final orthogonal output is then $Y = \tilde{Y}R^{-1}$. For further details about the training process with the orthogonalization layer, see Appendix F.

With some mathematical transitions (see Appendix E), the loss in Eq. 2 can be written in the following Rayleigh quotient form:

$$\mathcal{L}_\theta = \frac{2}{m^2v} \operatorname{Tr}\left(Y^T \sum_{p=1}^{v} L^{(p)} Y\right), \tag{4}$$

where $L^{(p)}$ is the $m \times m$ graph Laplacian of the $p$-th view.

One can observe that this loss is exactly the arithmetic mean version of the approximate joint diagonalization objective in Eq. 1 where each Laplacian is multiplied with the same weight. Our experiment in Appendix D.1 illustrates that our method effectively approximates the joint eigenvectors of the graph Laplacians.

The choice of affinity measure plays a crucial role in determining the quality of the generated representations. An appropriate affinity measure can enhance the ability of the model to capture the underlying relationships within the data, while an inadequate one may lead to poor representation quality and misinterpretation of the data structure. In Appendix B, we provide a discussion of the technique we employed to construct the affinity matrices.

**Generalizability and Scalability.** Once the framework is trained, it provides a mapping function $F_\theta$ that transforms each multi-view sample directly into its coordinates in the final unified representation, facilitating efficient generalization for new samples from the same distribution. Notably, all our experiments were conducted using test sets, which illustrates the method's generalizability. Additionally, the stochastic mini-batch training in SpecRaGE avoids computing the full Laplacian eigenvectors, enabling scalability for large datasets. For example, SpecRaGE processed the 1-million-sample InfiniteMNIST dataset (see Section 5.1) in about 15 minutes on a GTX-1080 Ti, whereas traditional graph Laplacian methods faced out-of-memory errors or much longer runtimes. The overall running time complexity of SpecRaGE is $O(n(k^2 + mv))$, where $k$ is the output dimension, $v$ is the number of views, and $m$ is the batch size. Since $k$, $v$, and $m$ are typically much smaller than $n$ and independent of $n$, our method exhibits near-linear time complexity. In Appendix C, we present a full time complexity analysis.

### 4.3 ROBUST META-LEARNING FUSION

**View-specific Representations.** As described above, the loss function in 2 operates on a fused representation $Y \in \mathbb{R}^{m \times k}$, derived from a mini-batch of $m$ multi-view samples. To construct this fused representation, we first need to generate intermediate representations for each individual view. The primary goal of these representations is to embed numeric and categorical features into a common space and ensure that all views are of the same size. To extract the view-specific representations, we introduce an individual neural network, $g_\theta^{(p)} : \mathbb{R}^{d_p} \to \mathbb{R}^k$ (e.g., an MLP), for each view. Specifically, given a multi-view input $\hat{x}_i = \left( x_i^{(1)}, x_i^{(2)}, \ldots, x_i^{(v)} \right)$, the intermediate representations are $y_i^{(1)}, y_i^{(2)}, \ldots, y_i^{(v)}$, where $y_i^{(p)} = g_\theta^{(p)} \left( x_i^{(p)} \right)$. For a batch of $m$ multi-view samples, we obtain $v$ matrices $Y^{(1)}, Y^{(2)}, \ldots, Y^{(v)}$, where $Y^{(p)}$ is an $m \times k$ matrix of the outputs, and its $i$-th row corresponds to $y_i^{(p)}$.

Merging the view-specific representations $Y^{(1)}, Y^{(2)}, \ldots, Y^{(v)}$ into a unified representation $Y$ presents a significant challenge. A key difficulty in this process arises from the potential presence of contaminated samples in some views, such as outliers or noisy data. In such cases, it is desirable to give less weight to the low-quality views in order to obtain a more accurate representation. Specifically, we need an approach that can evaluate the quality of each view directly from the data and determine its degree of contribution accordingly.

**Meta-learning for Fusion.** To dynamically weight views based on their quality, we introduce another neural network model that takes a multi-view sample as input and predicts a weights vector $w_i$. This model functions as a meta-learning model as it learns how to adaptively weight the views based on the data, optimizing the fusion process. The size of the weight vector corresponds to the number of views, and each entry in the vector indicates the importance or contribution of the corresponding view. We apply a softmax on the obtained vector to ensure that the vector sums up to one. The overall process is as follows: First, the view-specific representations $y_i^{(1)}, y_i^{(2)}, \ldots, y_i^{(v)}$ are obtained from the multi-view sample $\hat{x}_i$. Then, the concatenated $\hat{x}_i$ is passed through the meta-learning model, generating a weights vector $w_i$. Subsequently, to get the fused representation $y_i$, the following weighted sum is performed: $y_i = \sum_{p=1}^{v} w_i^{(p)} \cdot y_i^{(p)}$. Section 5.3 demonstrates that this fusion approach not only achieves state-of-the-art performance on standard multi-view benchmarks but also significantly outperforms existing methods in handling outliers and noisy views.

To provide meaningful feedback to the meta-learning model, we integrate the view-specific contributions into the loss function. For a batch of size $m$, weight vectors $w_1, w_2, \ldots, w_m$ are first generated using the meta-learning model. We then compute the mean of these weight vectors, $\bar{w}$, to capture a global representation of the batch's characteristics. This mean weight vector is embedded into the loss function, guiding the meta-learning model in producing appropriate weights for each view. The resulting loss function is defined as:

$$\mathcal{L}_\theta = \frac{2}{m^2 v} \text{Tr} \left( Y^T \sum_{p=1}^{v} \bar{w}^{(p)} L^{(p)} Y \right), \tag{5}$$

where $\bar{w}^{(p)}$ denotes the mean contribution of the $p$-th view.

## 5 EXPERIMENTS

### 5.1 EXPERIMENTAL SETTINGS

**Datasets.** We assess the performance of SpecRaGE using five well-studied multi-view datasets. Our selection prioritizes datasets that exhibit diversity in the types and number of views, as well as the number of classes, as depicted in Table 3 in Appendix A. The datasets are listed as follows: (1) **BDGP** (Cai et al., 2012) contains 2500 images of Drosophila embryos divided into five categories with two extracted features. One view has 1750-dimensional visual features, and the other view has 79-dimensional textual features. (2) **Reuters** is a multilingual dataset comprising more than 11,000

articles across six categories and five languages: English, French, German, Italian, and Spanish. We used a subset of this dataset containing 18,758 samples for our analysis. (3) **Caltech20** is a subset of 2386 examples derived from the object recognition dataset (Fei-Fei et al., 2004), which comprises 20 classes viewed from six different perspectives. The dataset encompasses various features such as Gabor features, wavelet moments, CENTRIST features, histogram of oriented gradients, GIST features, and local binary patterns. (4) **Handwritten** (Asuncion & Newman, 2007) contains 2,000 digital images of handwritten numerals ranging from 0 to 9. This dataset employs two types of descriptors: a 240-dimensional pixel average within $2 \times 3$ windows, and a 216-dimensional profile correlations method, serving as two distinct viewpoints. (5) **InfiniteMNIST** is a large-scale variant of MNIST (LeCun et al., 1998), with 1 million samples. The first view contains the original images, while the second adds Gaussian noise ($\sigma = 0.2$) to images from the same class, as in (Trosten et al., 2023). Created similarly to affNIST[1], this dataset applies small random affine transformations to expand the original data. More details about the different datasets can be found in Appendix A.

**Baselines.** We compared the performance of SpecRaGE with seven multi-view representation learning methods including two classic deep methods (DCCA (Andrew et al., 2013) and DCCAE (Wang et al., 2015)) and five state-of-the-art methods (MvSCN (Huang et al., 2019), MIB (Federici et al., 2020), Multi-VAE (Xu et al., 2021), AECoKM (Trosten et al., 2023), and MetaViewer (Wang et al., 2023)). DCCA and DCCAE are deep extensions of classic correlation analysis. MvSCN learns spectral embeddings for each view while aligning view-specific representations with an MSE objective. MIB uses information theory to separate shared and view-specific information. Multi-VAE disentangles visual representations into view-common and view-peculiar variables. AECoKM combines autoencoders with contrastive learning for view alignment. MetaViewer learns to fuse representations by observing reconstruction from unified representations to specific views and employs contrastive learning in its improved version. To ensure fairness, we ran each algorithm ten times on the aforementioned datasets using the same backbones, recording the mean and standard deviation of their performance. For clustering, we employed K-means, while Support Vector Machines (SVM) were used for classification. For alignment-based methods, we concatenated the view-specific representations before applying clustering and classification. More details on hyper-parameters and training are in Appendix F.

**Evaluation metrics.** For clustering, we employed three widely used metrics: Normalized Mutual Information (NMI), Accuracy (ACC), and Adjusted Rand Index (ARI). For classification, we used Accuracy, Precision, and F-score. Higher values of these metrics indicate superior performance.

## 5.2 REPRESENTATION EVALUATION

**Clustering Results.** In the clustering experiment, we aimed to evaluate SpecRaGE's ability to capture the underlying cluster structure of the data in comparison to the baseline methods. To achieve this, we applied the K-means algorithm to the final representations learned by each method. Table 1 summarizes the clustering results across the five datasets. SpecRaGE achieves top-performing results on most datasets and evaluation metrics. Fig. 2 provides an additional illustration of the efficacy of SpecRaGE in representing the data and capturing its inherent cluster structure.

**Classification Results.** In the classification experiment, we aimed to assess the quality of the learned representations by running a SVM on top of the final embeddings produced by each method. We chose to use SVM because it does not introduce any additional non-linear transformations to the feature spaces, ensuring that the comparison between different algorithms remains unbiased. SVM and linear classifiers are widely accepted as standard benchmarks for evaluating learned representations and embeddings (See, for example, (Chen et al., 2020; Federici et al., 2020; Bardes et al., 2021; Wang et al., 2023)).

Table 2 presents the classification results across the five datasets. Notably, SpecRaGE consistently achieves top performance in most datasets and evaluation metrics, demonstrating its effectiveness in generating discriminative representations for classification tasks.

---

[1]https://www.cs.toronto.edu/~tijmen/affNIST/

Table 1: Clustering results of all methods on four datasets. The best result in each row is shown in bold and the second-best is underlined.

| Datasets | Metrics | DCCA | DCCAE | MvSCN | MIB | Multi-VAE | AECoKM | MetaViewer | SpecRaGE |
|---|---|---|---|---|---|---|---|---|---|
| BDGP | ACC | 75.8 ±1.31 | 80.1 ±1.50 | 81.9 ±4.33 | 86.9 ±0.75 | 53.8 ±6.91 | 76.6 ±5.33 | 90.4 ±0.43 | **97.6** ±0.53 |
| | NMI | 67.9 ±1.21 | 73.2 ±1.04 | 76.9 ±4.15 | 80.9 ±0.91 | 29.9 ±6.54 | 68.7 ±5.89 | 86.3 ±1.05 | **93.4** ±0.83 |
| | ARI | 49.2 ±1.31 | 55.1 ±1.02 | 70.3 ±5.42 | 74.1 ±1.82 | 28.9 ±4.92 | 54.7 ±6.34 | 89.3 ±1.21 | **94.1** ±1.29 |
| Reuters | ACC | 47.9 ±0.93 | 42.0 ±1.23 | 49.5 ±2.50 | 49.5 ±1.10 | 36.6 ±2.34 | 26.5 ±0.52 | 47.8 ±0.08 | **56.7** ±3.22 |
| | NMI | 26.6 ±1.32 | 20.3 ±1.13 | 27.6 ±1.32 | 28.4 ±1.81 | 21.1 ±2.20 | 16.4 ±1.54 | 24.2 ±0.06 | **38.3** ±2.45 |
| | ARI | 12.7 ±1.44 | 8.5 ±1.06 | 23.1 ±1.12 | 24.4 ±1.23 | 12.4 ±2.34 | 10.1 ±2.54 | 19.19 ±0.81 | **29.6** ±2.64 |
| Caltech20 | ACC | 39.7 ±1.10 | 36.1 ±1.50 | 42.1 ±2.59 | 36.1 ±1.23 | 32.4 ±2.25 | 42.2 ±3.65 | 45.1 ±1.33 | **50.1** ±2.61 |
| | NMI | 51.2 ±1.21 | 52.4 ±1.71 | 53.7 ±3.49 | 47.5 ±1.55 | 46.2 ±2.82 | 57.6 ±0.18 | 60.9 ±2.15 | **66.0** ±1.72 |
| | ARI | 23.5 ±2.23 | 25.3 ±1.14 | 29.7 ±3.26 | 23.2 ±1.04 | 24.8 ±1.79 | 32.2 ±1.20 | 35.0 ±1.24 | **40.1** ±3.49 |
| Handwritten | ACC | 58.3 ±2.10 | 63.8 ±1.21 | 68.3 ±4.86 | 67.5 ±2.95 | 74.5 ±3.34 | 66.4 ±2.34 | 86.3 ±2.51 | **91.9** ±3.51 |
| | NMI | 70.2 ±1.69 | 75.2 ±1.34 | 70.9 ±2.90 | 64.7 ±2.18 | 70.0 ±3.65 | 70.3 ±2.70 | 78.9 ±1.44 | **86.5** ±1.74 |
| | ARI | 52.2 ±1.33 | 60.8 ±1.87 | 59.6 ±4.25 | 48.9 ±2.56 | 60.5 ±3.15 | 61.4 ±2.70 | 72.3 ±2.94 | **83.0** ±1.98 |
| InfiniteMNIST | ACC | 95.6 ±1.09 | 95.1 ±1.21 | **99.1** ±0.28 | 68.6 ±3.95 | 98.1 ±0.50 | **99.3** ±0.20 | 80.0 ±0.22 | **99.1** ±0.27 |
| | NMI | 90.0 ±1.39 | 88.9 ±1.41 | 97.7 ±1.54 | 67.2 ±3.15 | 96.0 ±1.20 | **98.1** ±0.42 | 72.3 ±0.15 | 97.5 ±0.54 |
| | ARI | 90.1 ±1.33 | 89.3 ±1.01 | 97.9 ±0.48 | 66.9 ±3.56 | 96.4 ±1.05 | **98.5** ±0.20 | 65.2 ±0.24 | 97.9 ±0.21 |

Table 2: Classification results of all methods on four datasets. The best result in each row is shown in bold and the second-best is underlined.

| Datasets | Metrics | DCCA | DCCAE | MvSCN | MIB | Multi-VAE | AECoKM | MetaViewer | SpecRaGE |
|---|---|---|---|---|---|---|---|---|---|
| BDGP | ACC | 98.40 ±0.81 | 98.65 ±0.26 | 98.76 ±0.10 | 90.56 ±1.55 | 88.87 ±2.54 | 98.92 ±0.21 | 98.00 ±0.11 | **99.01** ±0.30 |
| | F-score | 98.40 ±0.82 | 98.50 ±0.21 | 98.76 ±0.10 | 90.10 ±0.60 | 88.87 ±2.54 | 98.92 ±0.20 | 98.02 ±0.11 | **99.00** ±0.23 |
| | Precision | 98.42 ±0.61 | 98.63 ±0.30 | 98.02 ±0.15 | 89.92 ±1.32 | 89.07 ±2.43 | 99.01 ±0.20 | 98.59 ±0.10 | **99.01** ±0.20 |
| Reuters | ACC | 74.40 ±0.92 | 74.10 ±0.87 | 75.52 ±0.12 | 71.96 ±1.02 | 62.06 ±3.40 | 68.01 ±0.85 | 59.17 ±0.10 | **76.61** ±1.70 |
| | F-score | 74.50 ±1.10 | 74.21 ±0.90 | 75.51 ±0.12 | 70.78 ±0.91 | 59.60 ±3.95 | 65.21 ±0.81 | 51.19 ±0.09 | **76.60** ±1.77 |
| | Precision | 74.72 ±0.92 | 74.35 ±1.01 | 75.53 ±0.14 | 70.78 ±1.10 | 61.31 ±3.22 | 67.24 ±0.85 | 56.43 ±0.12 | **78.59** ±1.75 |
| Caltech20 | ACC | 72.60 ±0.51 | 72.58 ±0.64 | 86.54 ±0.35 | 73.52 ±2.41 | 87.73 ±0.63 | 93.38 ±1.07 | 92.16 ±0.05 | **95.42** ±0.92 |
| | F-score | 40.12 ±0.50 | 43.26 ±0.81 | 86.75 ±0.37 | 72.52 ±2.10 | 87.30 ±0.91 | 93.03 ±0.97 | 85.72 ±0.10 | **95.93** ±0.93 |
| | Precision | 46.30 ±0.44 | 60.66 ±0.69 | 85.32 ±0.33 | 73.25 ±1.93 | 88.90 ±1.60 | 93.69 ±1.26 | 90.68 ±0.15 | **95.44** ±0.90 |
| Handwritten | ACC | 88.25 ±0.91 | 90.01 ±0.45 | 96.21 ±0.21 | 96.01 ±1.04 | 95.36 ±0.95 | 96.91 ±0.65 | 97.75 ±0.21 | **97.80** ±1.27 |
| | F-score | 88.05 ±1.03 | 89.92 ±0.45 | 96.21 ±0.21 | 96.10 ±1.20 | 95.36 ±0.93 | 96.95 ±0.63 | 97.75 ±0.20 | **97.77** ±1.27 |
| | Precision | 89.20 ±0.85 | 90.48 ±0.56 | 96.24 ±0.21 | 96.03 ±0.90 | 95.38 ±0.92 | 96.91 ±0.34 | **97.90** ±0.19 | 97.80 ±1.26 |
| InfiniteMNIST | ACC | 97.21 ±0.12 | 97.60 ±0.41 | 99.12 ±0.10 | 95.52 ±0.11 | 98.75 ±0.02 | **99.50** ±0.09 | 95.71 ±0.10 | 99.35 ±0.04 |
| | F-score | 97.21 ±0.14 | 97.60 ±0.41 | 99.12 ±0.10 | 95.51 ±0.11 | 98.75 ±0.02 | **99.50** ±0.08 | 95.71 ±0.10 | 99.34 ±0.04 |
| | Precision | 97.21 ±0.12 | 97.60 ±0.41 | 99.12 ±0.10 | 95.56 ±0.11 | 98.78 ±0.03 | **99.51** ±0.09 | 95.74 ±0.15 | 99.35 ±0.04 |

**Visualization.** To further demonstrate the effectiveness of the learned unified representation, we utilize the $t$-SNE algorithm on the representation obtained from the validation set during the training. As depicted in Fig. 2, our method successfully separates the data into distinct clusters with increasing training epochs.

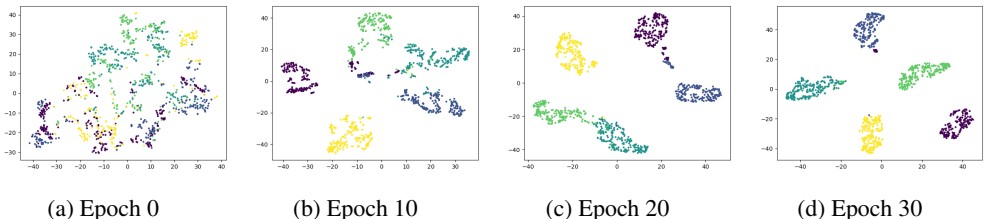

| (a) Epoch 0 | (b) Epoch 10 | (c) Epoch 20 | (d) Epoch 30 |

Figure 2: Visualization of the unified representation $Y$ during training on the BDGP dataset.

## 5.3 ROBUSTNESS TO CONTAMINATION EVALUATION

In real-world scenarios, data contamination is a pervasive challenge, whether in the form of noise or anomalous outliers, often introduced by faulty sensors, human errors, or external environmental factors. These types of contamination can severely degrade the performance of multi-view models, making robustness a critical capability. To evaluate SpecRaGE's robustness—a key differentiator of our method—we designed two contamination experiments: one targeting robustness to outliers and the other to Gaussian noise.

In the outlier experiment, global anomalies were randomly injected into one view, following the approach from (Han et al., 2022). For the noise experiment, Gaussian noise with $\sigma = 1.2$ was injected into a portion of the samples in one randomly selected view. We conducted both experiments at contamination ratios of 10%, 20%, 30%, and 40%.

These tests were performed on two real-world datasets and a synthetic 2-view version of the scikit-learn Blobs dataset [2]. We chose to include this simple synthetic 2D data in this experiment, as it allows us to more clearly distinguish the effect of the noise or outliers on the model's performance. In these experiments, we report the relative degradation percentage with respect to the uncontaminated baseline, where no contamination is injected.

As shown in Fig 3, SpecRaGE consistently outperforms other methods in terms of robustness, maintaining stable performance even under high levels of contamination. In both the outlier and noisy view experiments, SpecRaGE demonstrates significantly smaller relative degradation percentages across all datasets, even at the highest contamination ratios. This resilience to extreme contamination underscores the method's effectiveness in handling challenging real-world scenarios, where high levels of noise or outliers are often unavoidable.

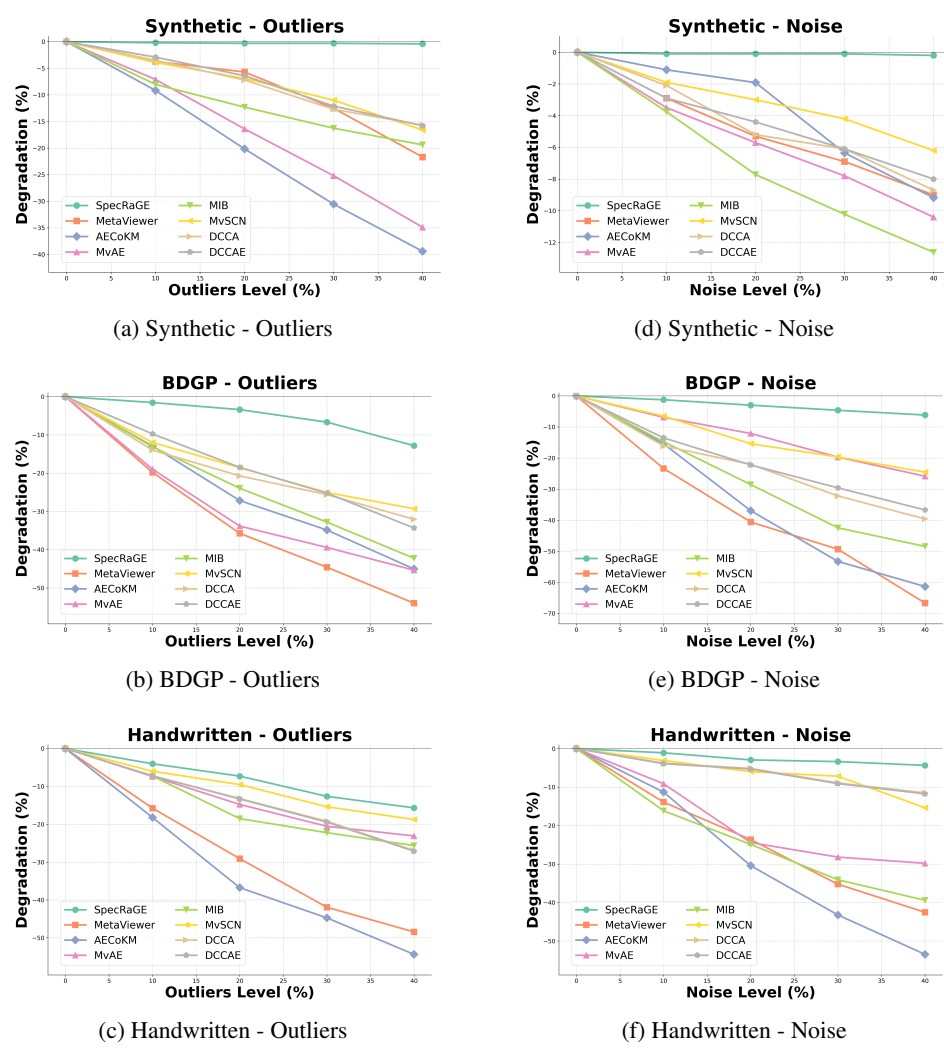

Figure 3: Performance decrease of methods with various levels of outliers (left column) and noise (right column) across different datasets. Each line represents a different method, showing the relative performance degradation (%) as the percentage of outliers or noise increases.

---

[2]https://scikit-learn.org/stable/modules/generated/sklearn.datasets.make_blobs.html

## 6 CONCLUSION

In this work, we introduced SpecRaGE, a novel fusion-based framework for multi-view spectral representation learning that effectively integrates graph Laplacian methods with deep learning techniques. By efficiently learning a parametric map to uncover joint eigenvectors from diverse graph Laplacians, SpecRaGE addresses the challenges of generalizability and scalability, enabling it to handle large datasets while generalizing to new samples. Moreover, SpecRaGE employs a dynamic fusion technique that enhances robustness against outliers and noise in contaminated multi-view data. Extensive experiments validate that SpecRaGE achieves state-of-the-art performance on standard multi-view benchmarks and significantly outperforms existing methods when faced with outliers and incomplete views. These results highlight SpecRaGE's potential to transform multi-view learning in practical applications where data quality is often compromised.

**Reproducibility Statement.**   A complete explanation of SpecRaGE's logic is provided in Section 4 and Algorithm 1. All key details for reproducibility, including the training methodology with the orthogonality constraint, network backbones for both the view-specific and meta-learning networks, hyperparameters, data splits, operating system, and hardware specifications, are available in Appendix F. Further discussion on the construction of affinity matrices is found in Appendix B.

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

## A    DATASETS CHARACTERISTICS

In Table 3, we provide additional information regarding the sample size, number of views, and dimensions of the various datasets.

Table 3: Characteristics of the datasets in our experiments

| Dataset | #Samples | #Classes | #Views | #Dimensions |
|---|---|---|---|---|
| BDGP | 2500 | 5 | 2 | 1750; 79 |
| Reuters | 18,758 | 6 | 5 | 21531; 24892; 34251; 15506; 11547 |
| Caltech20 | 2386 | 20 | 6 | 48; 40; 254; 1984; 512; 928 |
| Handwritten | 2000 | 10 | 2 | 240; 216 |
| InfiniteMNIST | 1,000,000 | 10 | 2 | 784;784 |

## B    SELF-SUPERVISED AFFINITY LEARNING.

To perform joint diagonalization of Laplacians, it is necessary to know how to construct an affinity matrix for each view. A widely used approach is to use a Gaussian kernel with a specific scale parameter $\sigma > 0$. For instance, for the $p$-th view, the affinity matrix is defined as follows:

$$W_{i,j}^{(p)} = \begin{cases} \exp\left(-\frac{\|x_i^{(p)} - x_j^{(p)}\|^2}{2\sigma^2}\right), & x_j^{(p)} \text{ is one of the } l \text{ nearest neighbors of } x_i^{(p)}, \\ 0, & \text{otherwise.} \end{cases} \tag{6}$$

However, Euclidean distance may offer a limited measure of similarity, particularly for high-dimensional data (Beyer et al., 1999; Aggarwal et al., 2001). Therefore, instead of directly computing the Euclidean distance between $x_i$ and $x_j$, we choose to train a SiameseNet (Koch et al., 2015) for each view $p$, denoted as $h_{\theta_{\text{siamese}}}^{(p)}$. This replaces the Euclidean distance $\|x_i^{(p)} - x_j^{(p)}\|^2$ in Eq. 6 with $\|z_i^{(p)} - z_j^{(p)}\|^2$, where $z_i^{(p)} = h_{\theta_{\text{siamese}}}^{(p)}\left(x_i^{(p)}\right)$.

Training a SiameseNet typically requires positive and negative pairs. When labels are provided, pairs from the same class are considered positive, while those from different classes are negative. In our unsupervised setup, we determine positive pairs based on small Euclidean distances between points $x_i^{(p)}$ and $x_j^{(p)}$, and negative pairs otherwise. Specifically, positive pairs are formed from the $l$ nearest neighbors of each point, while negative pairs are generated from points that are not included in the nearest neighbors set. The parameters $l$ and $\sigma$ are hyper-parameters, discussed in Appendix F.

The training of the SiameseNets serves as a pre-processing step. Once the SiameseNets are trained, we use them to construct batch affinity matrices for each view during the training. As in (Shaham et al., 2018), we empirically found (see the ablation study in Appendix D.2 for the experiments) that in various datasets, using SiameseNet for the affinities improves the quality of the representations.

## C    Time Complexity Analysis

---

**Algorithm 1** SpecRaGE

---

1: **Input:** Multi-view data $\hat{x}_1, \hat{x}_2, \ldots, \hat{x}_n$, output dimension $k$, batch size $m$, number of epochs $T$
2: **Output:** Fused representation $y_1, \ldots, y_n = Y \in \mathbb{R}^{n \times k}$ - the approximate joint eigenvectors.
3: **for** each epoch $t \in \{1, 2, \ldots, T\}$ **do**
4:     **for** each mini-batch of size $m$ **do**
5:         Obtain view-specific representations $Y^{(1)}, Y^{(2)}, \ldots, Y^{(v)}$
6:         Construct $m \times m$ affinity matrices $W^{(1)}, W^{(2)}, \ldots, W^{(v)}$
7:         Compute the corresponding graph Laplacians $L^{(1)}, L^{(2)}, \ldots, L^{(v)}$
8:         Generate $m$ weights vectors $w_1, w_2, \ldots, w_m$ using the meta-learning model
9:         Obtain the fused representation $\tilde{Y}$ using the weights vectors
10:         Apply orthogonality constraint using QR decomposition: $Y = \tilde{Y} R^{-1}$
11:         Compute the mean of the weights vectors $\bar{w}$
12:         Compute loss in 5
13:     **end for**
14: **end for**
15: Forward propagate $\hat{x}_1, \hat{x}_2, \ldots, \hat{x}_n$ and obtain $n$ the outputs $y_1, y_2, \ldots, y_n \in \mathbb{R}^k$

---

The time complexity of the algorithm in 1 can be analyzed as follows:

Given that:

- The size of the networks and the number of epochs are constant.
- Batch size is $m$.
- The number of views is $v$.
- The total number of samples is $n$.
- The output dimension is $k$.

The running time breakdown per batch is:

- line 5: $O(mv)$.
- lines 6-7: $O(m^2 v)$.
- line 8: $O(mv)$.
- line 9: $O(mv)$.
- line 10: $O(mk^2)$.
- line 11: $O(mv)$.
- line 12: $O(m^2 v)$.

The Overall complexity:

- Per batch: $O(4mv + mk^2 + 2m^2 v)$.
- Per epoch: $O(\frac{n}{m} \cdot (4mv + mk^2 + 2m^2 v)) = O(n(k^2 + v(2m + 4))) = O(n(k^2 + mv))$.

Now since $m$, $v$, and $k$ are much smaller than $n$, this method presents almost linear running time complexity.

## D    Further experiments

### D.1    Approximation of the joint eigenvectors

We begin by demonstrating that our unified representation approximates the joint eigenvectors. To show this approximation, we compute the Grassmann distance between the subspace of SpecRaGE's

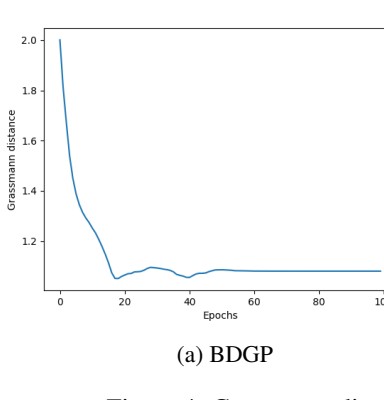 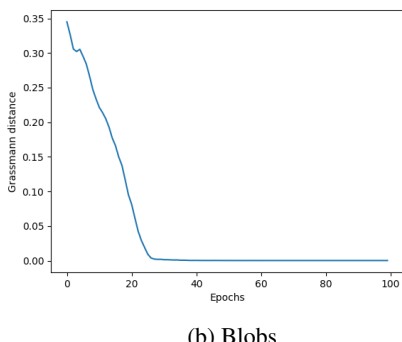

(a) BDGP                      (b) Blobs

Figure 4: Grassmann distance as a function of the number of epochs.

output and that of the true eigenvectors of the matrix $\sum_{p=1}^{v} L^{(p)}$, referred to as the joint eigenvectors. The squared Grassmann distance measures the sum of squared sines of the angles between two $k$-dimensional subspaces, yielding values within the $[0, k]$ range. To demonstrate this approximation, we used the BDGP dataset and the scikit-learn 2D Blobs dataset, where the second view is generated by applying a random rotation transformation to the first view.

In the BDGP dataset, our model has an output dimension of 5, so the maximum distance between the subspace of SpecRaGE's output and that of the joint eigenvectors is also 5. As shown in Fig. 4, this distance significantly decreases early in training and stabilizes around 0.9, indicating that SpecRaGE effectively approximates the joint eigenvectors. For the Blobs dataset, the Grassmann distance quickly approaches zero, indicating that the subspace of SpecRaGE output is extremely close, if not identical, to the subspace spanned by the exact true joint eigenvectors.

## D.2 Ablation study with SiameseNets

In Eq. 6 we show how we construct the Gaussian kernel affinities, essential for our primary joint diagonalization objective. To ensure these Gaussian kernels effectively encode the similarities between the data points, we opt to train a SiameseNet on each view. After the SiameseNets are trained, we compute the Euclidean distance on the Siamese's output instead of computing it on the feature space.

The goal of this experiment is to demonstrate that using the Siamese-based affinity significantly improves the quality of the final unified representation, implying that Siamese-based affinity captures more accurately the similarities between the points. In Table 4, it is evident that replacing the Euclidean distance computed on the feature space with the Euclidean distance computed on Siamese representations (referred to as Siamese distance) leads to significant improvements in clustering results. The "Euclidean distance" column presents the clustering results where Euclidean distance is utilized for each view's affinity ($W^{(p)}$), while the "Siamese distance" column showcases the results where Siamese distance is employed for each view's affinity.

Table 4: ACC clustering results with and without the SiameseNets.

| Datasets | Euclidean distance | Siamese distance |
|---|---|---|
| BDGP | 82.2 ±5.68 | 97.4±0.53 |
| Handwritten | 77.9 ±1.06 | 92.0 ±3.51 |
| InfiniteMNIST | 86.5 ±0.10 | 99.2 ±0.27 |

## E Equivalence of the loss functions

In Section 4.2, we claim that the loss function in Eq. 2 is equivalent to the joint-diagonalization loss in Eq. 4, that is:

$$\sum_{i,j=1}^{m} W_{i,j}\|y_i - y_j\|_2^2 = 2\operatorname{Tr}\left(Y^T \sum_{p=1}^{v} L^{(p)} Y\right)$$

Here, we provide a proof for this equation.

*Proof.* As demonstrated in (Belkin & Niyogi, 2003) the following relation holds:

$$\sum_{i,j=1}^{m} W_{i,j}\|y_i - y_j\|_2^2 = 2\operatorname{Tr}\left(Y^T L Y\right)$$

Therefore, we can derive the following expression:

$$\sum_{p=1}^{v}\sum_{i,j=1}^{m} W_{i,j}^{(p)}\|y_i - y_j\|_2^2 = 2\sum_{p=1}^{v}\operatorname{Tr}\left(Y^T L^{(p)} Y\right)$$

$$= 2\operatorname{Tr}\left(\sum_{p=1}^{v} Y^T L^{(p)} Y\right) = 2\operatorname{Tr}\left(Y^T \sum_{p=1}^{v} L^{(p)} Y\right)$$

□

## F  TECHNICAL DETAILS

For fairness, we run each of the compared algorithms ten times on the above datasets, recording both the mean and standard deviation of their performance. The same backbones are employed across all methods and datasets. Specifically, we used an MLP with hidden layers of sizes 1024, 1024, and 512 for all view-specific networks $g_\theta^{(p)}$ across all datasets and methods. The meta-learning model also uses an MLP backbone with three hidden layers, each containing 100 units.

Additionally, for certain datasets such as BDGP, InfiniteMNIST, and Reuters, we initially embedded the raw features of each view $\mathcal{X}^{(p)}$ using a pre-trained Autoencoder (AE) with hidden layers of sizes 512, 512, and 2048. This pre-trained AE is used to obtain a lower-dimensional input, typically containing less nuisance information, as shown in (Shaham et al., 2018).

The size of the output of our model ($k$), is determined by the number of categories of the data. In the case of coupled view methods like DCCA, DCCAE, and MIB, we present results based on the two best-performing views. For alignment-based methods, the final unified representations are produced through concatenation. Subsequently, clustering and classification tasks are conducted using K-means and SVM classifiers, respectively. Training typically took 35-50 epochs for each dataset.

**Training with Orthogonalization Layer.**  To train the model with the orthogonalization layer, we adopt a technique similar to that used in *SpectralNet* (Shaham et al., 2018). This technique employs a coordinate descent training approach comprising two main optimization steps: the "orthogonalization step" and the "gradient step". Each step involves processing a different mini-batch.

During the orthogonalization step, we forward a mini-batch through the model and compute the QR decomposition of the fused representation to update the weights of the orthogonalization layer. In the gradient step, we pass another mini-batch through the model and use the orthogonalization weights from the preceding orthogonalization step to orthogonal the output. Following this, we compute the loss and update the network weights via backpropagation, while keeping the weights of the orthogonalization layer unchanged.

After training the model, all weights, including those of the orthogonal layer, are fixed. Consistent with observations from *SpectralNet*, we empirically found that when employing large mini-batches, the orthogonalization layer can also approximately orthogonalize the output of other mini-batches towards the end of training. For instance, in the Blobs dataset, when a random batch of size 1024 passes through the model with the fixed orthogonalization layer (i.e., after training), the resulting

output $Y$, exhibits approximate orthogonality. This is evident in $Y^T Y$, where the average deviation of off-diagonal elements from 0 is merely 0.04.

**Hyper-parameters.** In Table 5, we provide a breakdown of the hyperparameters utilized for the various datasets. The #Neighbors parameter corresponds to the number of nearest neighbors ($l$) used for each data point in the Gaussian kernel, as outlined in Appendix B. Specifically, we select the same number of neighbors for each view $\mathcal{X}^{(p)}$, determined via hyperparameter tuning. The scale parameter ($\sigma$) is also utilized for the Gaussian kernel and was chosen for each view $\mathcal{X}^{(p)}$ as the median of the distances from any point in $\mathcal{X}^{(p)}$ to its $l$-nearest neighbors (across all points in the view), resulting in a global scale. The temperature parameter is applied in the softmax function used on the weight vector generated by the meta-learning model. Using a temperature greater than 1 in the softmax function helps to smooth the weight distribution across the views, reducing the absolute dominance of any single view. This is particularly useful when we want to ensure that all views contribute to the final representation, even if some might be slightly less informative.

The initial learning rate (LR) was uniformly set to $10^{-3}$ for all datasets, with a decay policy in place. This decay policy is contingent on monitoring the validation loss. If the validation loss fails to improve over 10 epochs, the LR is multiplied by 0.1. Furthermore, if the LR decreases to $10^{-8}$, training is terminated. Adam optimizer is used for training.

Table 5: Hyper-parameters.

| Hyper-params | BDGP | Reuters | Caltech20 | Handwritten | InfiniteMNIST |
|---|---|---|---|---|---|
| LR | $10^{-3}$ | $10^{-3}$ | $10^{-3}$ | $10^{-3}$ | $10^{-3}$ |
| Batch size | 1024 | 2048 | 1024 | 1024 | 1024 |
| #Neighbors ($l$) | 22 | 18 | 18 | 22 | 30 |
| scale ($\sigma$) | Global; Median | Global; Median | Global; Median | Global; Median | Global; Median |
| Softmax temperture | 250 | 500 | 250 | 250 | 250000 |

**Data Split.** For each dataset, we initially divide it into an 80% training set and a 20% testing set. Subsequently, for training, we further divide the training set into a 90% training subset and a 10% validation subset

**OS and Hardware.** The training procedures were executed on Rocky Linux 9.3, utilizing Nvidia GPUs including GeForce GTX 1080 Ti and A100 80GB PCIe.

