# OpenReview forum: "SpecRaGE: Robust and Generalizable Multi-view Spectral Representation Learning"
_ICLR.cc/2025/Conference — ICLR 2025 Conference Withdrawn Submission_

### Official Review · Reviewer_3oGr · 2024-10-28

**Soundness:** 2
**Presentation:** 2
**Contribution:** 1
**Rating:** 3
**Confidence:** 2

**Summary:**

This paper proposes a spectral representation learning framework that learns parameter maps to approximate the joint diagonalization of the graph Laplacian via neural networks, and designs a meta-learning fusion scheme that dynamically adapts to the quality of the data. This paper aims to improve the scalability, generalization and robustness of spectral representation learning.

**Strengths:**

- The paper is well organized, easy to follow. Problem statement is clear.
- Experiments demonstrate the effectiveness of SpecRage in the face of noisy and outlier data.

**Weaknesses:**

- Lack of Novelty: All proposed components are not new, and many similar frameworks has been proposed, such as [1].
- Regarding the module of meta-learning fusion, SpecRage simply using learnable weights, there are many similar works.
- The SpecRage lacks ablation experiments to verify the effectiveness of each module.

[1]ZhenyuHuang,JoeyTianyiZhou,XiPeng,ChangqingZhang,HongyuanZhu,andJianchengLv.
  Multi-viewspectralclusteringnetwork. InIJCAI,pp. 4,2019.

**Questions:**

In MvSCN, the networks used to learn the affinity matrix W share parameters. While in SpecRage, it's not shared, so what's the benefit?

---

### Official Review · Reviewer_u4TR · 2024-10-31

**Soundness:** 2
**Presentation:** 2
**Contribution:** 1
**Rating:** 3
**Confidence:** 5

**Summary:**

This paper integrates graph Laplacian methods and deep learning techniques to effectively process multi-view data, introducing a multi-view representation learning method called SpecRaGE. This method tries to address challenges related to generality, scalability, and robustness to contaminated views. The experimental results verify the effectiveness of the SpecRaGE.

**Strengths:**

1. This paper is easy to follow.
2. This paper proposed a method that integrates the strengths of graph Laplacian methods with the power of deep learning.
3. The experiments demonstrate the effectiveness of the proposed method.

**Weaknesses:**

1. This paper appears to be an incremental contribution, as it primarily combines graph Laplacian methods with deep learning techniques.

2. The authors claim to address generalization and scalability challenges; however, the paper lacks substantial contributions in these areas. 1) The generalization is achieved by using a neural mapping function for new samples from the same distribution, a capability shared by most deep clustering methods. 2) The scalability is handled through stochastic mini-batch training, which was already introduced in [SpectralNet: Spectral Clustering using Deep Neural Networks, ICLR 2018].

3. In lines 296-301, the concatenated  xˆ_i  is fed into the meta-learning model to generate a weight vector w_i. However, the implementation of the meta-learning approach is unclear. Despite claiming a novel fusion-based framework, the paper does not convincingly demonstrate a meaningful contribution to robust meta-learning fusion. The generated weight vector is used to fuse multi-view representations, but this approach is widely used. What are the specific contributions this paper offers?

4. The paper states that SpecRaGE utilizes dynamic fusion technology to improve robustness against outliers and noise in contaminated multi-view data. While empirical results support this claim, a theoretical explanation would strengthen it considerably.

5. Robustness entails mitigating noise, as demonstrated by robust principal component analysis (RPCA), where X=A+E, with X as the original data, A as the clean data, and E as the noise component. For instance, a partially occluded face introduces localized noise. Assigning weights to a low-quality view affects important, noise-free features, but this weighting process reduces rather than eliminates noise impact.

6. SpecRaGE employs joint diagonalization of Laplacians. A visualization of the diagonalization structure would help readers intuitively grasp this concept.

**Questions:**

Please see weaknesses.

---

### Official Review · Reviewer_k5ae · 2024-11-01

**Soundness:** 2
**Presentation:** 2
**Contribution:** 1
**Rating:** 3
**Confidence:** 4

**Summary:**

The paper introduces SpecRAGE, a framework combining graph Laplacian-based methods with deep learning for robust and scalable multi-view representation learning (MvRL). SpecRAGE addresses issues in traditional MvRL approaches with noise and outliers by using a neural network-based fusion module that adapts dynamically to data quality.

**Strengths:**

1.	SpecRAGE’s integration of graph Laplacian methods with deep learning for robust multi-view representation learning is a creative approach that addresses key limitations in handling noisy data.
2.	The paper is clearly structured, with thorough experimental validation showing SpecRAGE’s effectiveness and performance advantages over state-of-the-art methods.

**Weaknesses:**

1.	Formula specification problem: Using the \mathcal{X} to represent the eigenmatrix does not seem to be a canonical way of writing at Problem Statement. Is \hat{x} usually used to represent x for prediction? Y denotes the categories of data, it doesn't seem particularly appropriate to use View-specific Representations as a definition. The author's variables are unusual, such as using lowercase v for the total number of views and lowercase q for the index of views.
2.	SpecRaGE’s approximate joint diagonalization may introduce cumulative errors when Laplacians are non-commutative, affecting feature accuracy. Whether the author has considered this question.
3.	The dynamic weighting module’s stability in high-noise conditions is uncertain and needs testing under higher noise ratios.
4.	The basis of weight generation by meta-learning modules is not explained in detail, especially when the data quality difference is significant, whether the weight generation is reliable and how to avoid misleading weight allocation is worthy of further discussion. We didn't see the definition of the weight vector w, which seems to lack some description at Meta-learning for Fusion.
5.	The baseline selection of the experiment is too old, I think ICLR submission should choose the more latest algorithms of this year as a comparison, such as APADC, CPSPAN and so on.
6.	The ablation experiment is not sufficient, whether meta-model weighting is needed, and whether view fusion is necessary.

**Questions:**

Please refer to above-mentioned weaknesses.

---

### Official Review · Reviewer_BJVx · 2024-11-03

**Soundness:** 2
**Presentation:** 3
**Contribution:** 2
**Rating:** 3
**Confidence:** 4

**Summary:**

This paper introduces a fusion based multi-view spectral representation learning framework, SpecRaGE, which aims to solve the shortcomings of existing multi-view representation learning methods in generalization, scalability and robustness to noise and outliers.

**Strengths:**

1. SpecRaGE is a novel fusion-based multi-view spectral representation learning framework that effectively integrates graph laplacian methods with deep learning techniques.
2. Solves generalization and scalability challenges and is able to handle large data sets and generalize new samples.
3. Dynamic fusion technology is used to enhance the robustness of outliers and noise in contaminated multi-view data.
4. Achieved state-of-the-art performance in standard multi-view benchmarks, significantly outperforming existing methods when dealing with outliers and incomplete views.

**Weaknesses:**

1. There is no explanation of the newly introduced input character symbols in the Figure 1, and the readability of the graph is not good.
2. Generally speaking, for clustering and classification tasks, the size of the dataset often has a significant impact on the performance of the algorithm. In the experiments of this paper, we were surprised to find that its performance on a dataset InfiniteMNIST with a scale of one million was similar to that of a dataset BDGP with a scale of 2k. The author did not provide an analysis of this phenomenon. Besides, The experimental data is not the real contaminated data, and there is a certain gap with the real situation. Therefore, the effectiveness is weak.
3. The selected comparative experiment does not have representative significance, and the depth method was chosen in 2013 and 2015. It is unclear the significance of choosing these two outdated methods. Serious lack of experimental analysis, simply presenting the experimental results.

**Questions:**

1. Why is the Gaussian noise selected to be 1.2 instead of other values in the Line 433？
2. Is the affinity matrix learnable? If so, what is the learning strategy?
3. In the Section “Meta-learning for Fusion”, are the weights of each view simply arithmetic averages? If so, have other forms of averages been considered?

---

### Official Review · Reviewer_M2BJ · 2024-11-03

**Soundness:** 3
**Presentation:** 3
**Contribution:** 2
**Rating:** 5
**Confidence:** 5

**Summary:**

This paper presents an approach for multi-view representation learning. The primary motivation behind this study is to employ the joint diagonalization of graph Laplacians to help fuse the information from multi-view data points into a unified framework for obtaining latent representations. Technically, the authors discussed three challenges: generalizability for unseen (newly obtained) data, scalability for large-scale data sets, and robustness against outliers and noisy views. To address these limitations, deep learning method is introduced into the framework of joint graph Laplacian representation learning from all views.

Technically, the main and fundamental techniques in this paper are constructed on the following two works, namely, the SpectralNet (Shaham et al., 2018) and the approximate joint diagonalization of Laplacians. Based on this justification, in my opinion, the main contribution of this work lies in that the authors utilized the SpectralNet and the joint diagonalization together in the work setting of multi-view graph-embedding. Technically, beyond on the average of the different Laplacians, they developed a metal learning model in terms of MLP to obtain a linearly-weighted Laplacian for approximate joint diagonalization. In this way, they associated it to the third issue of robustness against outliers and noisy views.

**Strengths:**

The authors presented an approach for multi-view representation learning. In this framework, they combined together the advantages of the SpectralNet and the tricks of the graph Laplacian embedding for multi-view representation learning. First, the approximate joint diagonalization of Laplacians is addressed mathematically, which opens the door to graph embedding from single view to multiple views. Second, technically, they developed a metal learning model in terms of MLP to weight the single-view Laplacians. The comparative experiments indicate the validity of the proposed framework. Following these, the paper is well organized, the technical details are well described, and the writing is good for understanding. The analyses are also insightful.  The experiments are reported to support their main declarations.

**Weaknesses:**

However, the innovation of the framework developed in this study is not distinctly significant. Technically, the main and fundamental techniques are constructed on the following two works, namely, the SpectralNet (Shaham et al., 2018) and the approximate joint diagonalization of Laplacians. First, the SpectralNet (Shaham et al., 2018) is constructed by integrating deep learning into the graph-Laplacian embedding, which solves the fundamental problem of generalizability for unseen data (namely, the task of out-of-sample embedding). This is the primary limitation in traditional graph-Laplacian embedding, and well known in the field of manifold learning.  Different from most traditional manifold learning methods, here it yields a mapping recorded in the neural network for unseen data. In addition, with the tricks by using mini-batch training in deep learning, it is natural to extend the scale of data points to large-scale. Second, as also declared by the authors, the work in (Eynard et al., 2012; 2015) indicates that it is possible to obtain an approximate joint diagonalization of different Laplacians. As a whole, indeed, the work has novel points. But, based on the above observations, in my opinion, the innovation of the framework in this paper is somewhat limited to some degree.

Other issues:

1.	The framework is developed on the existing SpectralNet from single view to multi-views. As an iterative approximation for intrinsic embedding in the work setting of deep learning, there is a gap in SpectralNet between those embeddings obtained from the graph Laplacian constructed on mini-batches and those obtained on the whole (global) dataset. That is, algebraically, the orthogonalization on a series of small matrices will not equivalent to that on a large matrix. Although this processing is recorded in the network by training, and also although the SpectralNet can indeed return acceptable results for date clustering and classification, this gap could be enlarged when performing multi-view data. As a result, the learned embedding may be far from that obtained strictly by traditional manifold learning approaches. For example, there is a well-known dataset, which includes 400 images taken from a teapot with different viewpoints along a circle (K.Q. Weinberger and L.K. Saul, Unsupervised Learning of Image Manifolds by Semidefinite Programming, CVPR, pp. 988-995, 2004). The low-dimensional intrinsic embeddings will render a circle on 2D plane (strictly speaking, it is an one-dimensional intrinsic structure hidden in the data) . Within the framework constructed by this work, it is hoped that the first view of the original images can output a circle by the proposed SpecRaGE, and that the second view of the images with Gaussian noise could also output a circle. There may be other similar or more illustrative examples by performing such demonstrations for validation.

2.	Figure 2 illustrates an example of visualization. Its focus is on the effectiveness of iterative processes for deep learning and also for the goodness of data embedding. It is true. However, in my opinion, what is more important is to visualize，simultaneously，the results obtained by the deep learning and those obtained by traditional manifold learning methods that are constructed on all of the data points (for example, obtained from one view).
3.	Technically, it is necessary, theoretically or empirically, to demonstrate the power of the joint graph Laplacian by adding together (or in a linearly weighting way) all of the single-view Laplacian matrices. Mathematically, its algebraic meaning is actually not clear.

4.	The authors declared that the developed framework considers the robustness against outliers and noisy views. The experiments in subsection 5.3 are rich on two cases (outlier and noise). However, both of these cases are added manually by algorithm. Thus, it is nice to conduct the experimental evaluations on the data sets from real-world applications. What is more, and how about the situations if we employ the proposed SpecRaGE to multi-modal datasets from real-world applications, for example, one is related to image, while the other is associated to text or other. Note that, obtaining the joint embeddings (representations) with high quality is the key to perform the downstream pattern recognition tasks.

5.	Compared with the SOTA methods (like AECoKM and MetaViewer), the performances of the proposed SpecRaGE do not show significant enhancement. Thus, it is necessary to discuss the advantages over the SOTAs in training/inferring time, model complexity, or/and computation complexity, and others.

6.	The framework as a whole is not end-to-end. There are some tricks that could affect the final embeddings. Note that, although the experiments are conducted on the tasks of data clustering and classification, the more important job lies in the joint embedding. Comments are needed here for discussions or for carrying out future works.

**Questions:**

1.	The authors declared that some of the modern deep learning-based MvRL approaches rely on alignment, and their framework of joint diagonalization proposed in this paper can help avoiding the need for alignment between views, which can successfully circumvent the challenges associated with rigid view alignment. However, in the task formulation, all the data points collected from all of the views are strictly aligned in a multi-view input (see the $\hat x_i$). This confuses me a lot.

2.	The experiment in Appendix D.1 demonstrates that the proposed method can approximate the joint eigenvectors of the graph Laplacians. More details are needed. Specifically, how and from what to estimate the subspace of SpecRaGE’s output, from the eigenvectors obtained via the last mini-batch, or estimating it by the embeddings of all of the multi-view training data points after the network is well trained. In addition, why not calculated from the weighted total Laplacian?

3.	Technically, suppose there are totally $v$ views, each of them can yield a single low-dimensional embedding via the traditional manifold learning algorithm or the SpectralNet for single view with $n$ data points. With the averaged graph Laplacians, we can also obtain an embedding by performing the traditional manifold learning algorithm, denoting it here by $U$ for $n$ multi-view data points. So, if we can say or not that all of the $v$ embeddings are mapped in way of deep learning like SpecRaGE into a latent space spined (defined) by $U$ (at most there exist a gap with one linear transformation) for fusion representation?

4.	In the SpecRaGE, a meta learning is developed to evaluate the importance of each view to the multi-view fusion. How about the situation if it is replaced by an attention mechanism, or equivalently, it is just an attention mechanism across views？

---

### Note · Authors · 2024-11-25

**Comment:**

We thank the reviewers for their constructive comments and valuable feedback. Based on the current comments, we have decided to withdraw this version of the manuscript to develop a stronger and more comprehensive submission. Nevertheless, we would like to address some of the repeated comments raised by the reviewers for clarity and to outline directions for improvement.

1. **Meta-learning Module**
   To address the comments raised by reviewers M2BJ, k5ae, u4TR, and 3oGr, we provide a more detailed explanation of the Meta-learning fusion module. We will first outline the motivation and logic behind this approach, followed by a deeper dive into the technical aspects.

   - **Logic and Core Motivation**
     In multi-view representation learning, the quality of individual views can vary significantly due to noise, outliers, or data corruption. If views are treated equally, through methods like simple arithmetic averaging or uniform weighting, it can lead to suboptimal or even misleading representations. For instance, if one view is heavily contaminated, its corresponding graph Laplacian becomes distorted, introducing significant errors when summed with other Laplacians. As a result, using a simple average or unweighted fusion can result in representations that fail to capture the true underlying structure of the data.

   - **How the Model Learns to Weight Views**
     The fusion mechanism is inherently tied to the loss function, which penalizes poor representations. If a view's graph Laplacian is significantly damaged due to contamination, its contribution to the weighted sum of Laplacians can disrupt the optimization process. Through training, the meta-learning model adapts by reducing the weight assigned to such contaminated views, effectively minimizing their impact on the fused representation. This adaptation allows the model to "learn" which views to trust more in varying data conditions.

     For example, a clean and informative view will have a well-structured graph Laplacian, which contributes positively to the learning process by providing stable and meaningful gradients. In contrast, a contaminated view produces a noisy Laplacian, which can disrupt the optimization process by introducing unstable gradients or conflicting signals. This makes it harder for the model to converge effectively. The meta-learning module "identifies" such problematic views and assigns them lower weights, ensuring that the optimization process prioritizes more reliable views.

   In the next version of this manuscript, we will include an ablation study that compares our Meta-learning fusion module with several common fusion techniques, including simple averaging, concatenation, a learned linear layer, and an attention mechanism across views.

2. **Theoretical Analysis**
   As requested by reviewers u4TR, k5ae, and M2BJ, we provide a more detailed theoretical explanation of the concept of approximate joint diagonalization of Laplacians (AJDL). Following prior works [1], [2], the AJDL objective can be formulated as:

   $\min_{V} \sum_{i=1}^{v} \text{off}(V^\top L_i V) \quad \text{subject to } V^\top V = I$
   where $\text{off}(X) = ||X - \text{diag}(X)||^2_F$.

   When seeking $k$ joint null eigenvectors, this objective is equivalent to minimizing $\sum_{i=1}^{v}||L_i V||^2_F$. The intuition behind this equivalence is that $\sum_{i=1}^{v}||L_i V||_F^2$ captures both the diagonal and off-diagonal elements of $V^\top L_i V$. For true null eigenvectors, both the diagonal and off-diagonal elements should approach zero. Consequently, minimizing this energy simultaneously achieves joint diagonalization and identifies common null eigenspaces across all views.

   Since $\sum_{i=1}^{v}||L_i V||^2_F = \text{Tr}\left(V^\top \left(\sum_{i=1}^{v}L_i^\top L_i \right) V\right)$, the problem of finding joint eigenvectors becomes the problem of finding eigenvectors of the matrix $\sum_{i=1}^{v}L_i^\top L_i$. This matrix aggregates information from all the Laplacians $L_1, \dots, L_v$, and can be thought of as a combined representation that reflects their collective influence. In this work, we replaced the aggregation operator with a weighted sum of the Laplacians, as discussed in Section 4.3 and supported in [1] and [2].

   Even when Laplacians do not have exactly $k$ zero eigenvalues, our objective remains effective as it finds vectors that simultaneously minimize the eigenvalues across all views, effectively approximating joint eigenvectors.

3. **Scalability and Generalization**
   The reviewers BJVx and u4TR raised concerns regarding the scalability and generalizability of SpecRaGE. To address these points, we clarify that SpecRaGE is primarily a **representation learning method**. The focus of this work is on learning meaningful data representations, with clustering and classification tasks being downstream applications performed on relatively small test sets. Therefore, the time measurements reported in this paper correspond to the process of learning and obtaining the representations, not the downstream tasks themselves. This ensures that scalability and generalization are evaluated based on the efficiency of the representation learning process, rather than the specific performance on clustering or classification tasks.

   Additionally, we acknowledge that generalizability is a feature shared by many deep clustering methods. However, it is not inherently present in traditional "classic" graph Laplacian methods. By incorporating deep neural networks, we enable our graph Laplacian method to achieve this capability. It is important to note that we do not claim to "invent" the concept of generalization but rather to apply it to a domain where it was previously not achievable -- which is part of this work's novelty.

4. **Real-world Contaminated Data**
   To address the concerns raised by reviewers M2BJ and BJVx regarding "real-world" contamination, we acknowledge that the most commonly used benchmarks in the field of Multi-view Representation Learning (MvRL) are somewhat limited. These benchmarks typically consist of small, outdated, and relatively simple datasets that do not adequately reflect the complexities of real-world scenarios. In future work, we plan to explore more relevant datasets that incorporate inherent contamination to better assess the robustness and effectiveness of SpecRaGE in handling such challenges.



[1] Davide Eynard, Klaus Glashoff, Michael M Bronstein, and Alexander M Bronstein. Multimodal
diffusion geometry by joint diagonalization of laplacians. arXiv preprint arXiv:1209.2295, 2012.

[2] Davide Eynard, Artiom Kovnatsky, Michael M Bronstein, Klaus Glashoff, and Alexander M Bronstein.
Multimodal manifold analysis by simultaneous diagonalization of laplacians. IEEE transactions
on pattern analysis and machine intelligence, 37(12):2505–2517, 2015.

**Withdrawal Confirmation:**

I have read and agree with the venue's withdrawal policy on behalf of myself and my co-authors.